⊙ | **Open Peer Review** | Environmental Microbiology | Research Article
# *In situ* cell division and mortality rates of SAR11, SAR86, *Bacteroidetes*, and *Aurantivirga* during phytoplankton blooms reveal differences in population controls

Jan D. Brüwer,[1] Luis H. Orellana,[1] Chandni Sidhu,[1] Helena C. L. Klip,[2] Cédric L. Meunier,[2] Maarten Boersma,[2,3] Karen H. Wiltshire,[2,4] Rudolf Amann,[1] Bernhard M. Fuchs[1]

**ABSTRACT**   Net growth of microbial populations, that is, changes in abundances over time, can be studied using 16S rRNA fluorescence *in situ* hybridization (FISH). However, this approach does not differentiate between mortality and cell division rates. We used FISH-based image cytometry in combination with dilution culture experiments to study net growth, cell division, and mortality rates of four bacterial taxa over two distinct phytoplankton blooms: the oligotrophs SAR11 and SAR86, and the copiotrophic phylum *Bacteroidetes*, and its genus *Aurantivirga*. Cell volumes, ribosome content, and frequency of dividing cells (FDC) co-varied over time. Among the three, FDC was the most suitable predictor to calculate cell division rates for the selected taxa. The FDC-derived cell division rates for SAR86 of up to 0.8/day and *Aurantivirga* of up to 1.9/day differed, as expected for oligotrophs and copiotrophs. Surprisingly, SAR11 also reached high cell division rates of up to 1.9/day, even before the onset of phytoplankton blooms. For all four taxonomic groups, the abundance-derived net growth (−0.6 to 0.5/day) was about an order of magnitude lower than the cell division rates. Consequently, mortality rates were comparably high to cell division rates, indicating that about 90% of bacterial production is recycled without apparent time lag within 1 day. Our study shows that determining taxon-specific cell division rates complements omics-based tools and provides unprecedented clues on individual bacterial growth strategies including bottom–up and top–down controls.

**IMPORTANCE**   The growth of a microbial population is often calculated from their numerical abundance over time. However, this does not take cell division and mortality rates into account, which are important for deriving ecological processes like bottom–up and top–down control. In this study, we determined growth by numerical abundance and calibrated microscopy-based methods to determine the frequency of dividing cells and subsequently calculate taxon-specific cell division rates *in situ*. The cell division and mortality rates of two oligotrophic (SAR11 and SAR86) and two copiotrophic (*Bacteroidetes* and *Aurantivirga*) taxa during two spring phytoplankton blooms showed a tight coupling for all four taxa throughout the blooms without any temporal offset. Unexpectedly, SAR11 showed high cell division rates days before the bloom while cell abundances remained constant, which is indicative of strong top–down control. Microscopy remains the method of choice to understand ecological processes like top–down and bottom–up control on a cellular level.

**KEYWORDS**   growth, cell division, mortality, copiotrophy, oligotrophy, spring bloom, image cytometry

Address correspondence to Bernhard M. Fuchs, bfuchs@mpi-bremen.de.

The authors declare no conflict of interest.

See the funding table on p. 15.

Growth is an important ecological trait that reflects the success and activity of microbes in a given environment. Often, changes in cell numbers are referred to as apparent or net growth. But this does not take into account that net growth is the sum of cell division and mortality rates. While the net growth of microbial taxa is often reported, little is known about the associated cell division and mortality rates and the temporal coupling of the latter two. We studied two spring phytoplankton blooms when a de-coupling of cell division and mortality rates can be expected. The initial phase of phytoplankton blooms is often characterized by a dynamic substrate-driven succession of bacterial taxa that is fueled by the release of carbon-rich algal polysaccharides and other organic matter (1, 2). They promote high abundances of copiotrophic clades, which characteristically react quickly to substrate pulses, in a context with initially low mortality. It has been argued that copiotrophic taxa thereby outcompete the slow-growing oligotrophs (3). For example, many taxa of the phylum *Bacteroidetes* are stereotypic copiotrophs with sizable genomes of up to 6 Mbp (4, 5). Owing to their fast growth rates (2.2 to 5.1/day) (6, 7), they can rapidly grow to high abundances during phytoplankton blooms (2, 8). *Aurantivirga* is such a representative genus of *Bacteroidetes*, which recur and are highly abundant during and after phytoplankton blooms in the North Sea (5, 9).

Oligotrophic taxa commonly have small genomes, which provide a more limited capability to react to environmental changes (10). They have little plasticity in their cell division rates and cell volumes and generally show slow cell division rates (<1/day) (11, 12). The well-studied oligotrophic SAR11 clade (13, 14), which thrives in nutrient-depleted waters, accounts for about a third of all the bacteria in surface ocean waters (15–17). Its ~1.3 Mbp genome is among the smallest of all known free-living bacteria (15) and its cultured representative *Pelagibacter ubique* is characterized by slow cell division rates in the laboratory (<0.5/day) (13, 14). Similarly, the gammaproteobacterial SAR86 clade also represents another group of ubiquitous and abundant oligotrophs in surface ocean water (18, 19), which has thus far evaded cultivation. Members of this clade have small genomes ranging from ~1.2–1.7 Mbp (18) and have been reported to be slow growing (~0.5/day) (20).

Here, we determine and compare the *in situ* cell division, net growth, and mortality rates of four well-characterized oligotrophic and copiotrophic taxonomic groups over the course of two spring phytoplankton blooms. While net growth rates can be calculated from changes in the number of individuals over time, determining the net growth of a particular microbial population in a complex sample is inherently difficult due to the lack of unique morphological features of the unicellular organisms. Fluorescence *in situ* hybridization (FISH) allows the identification and detection of individual taxonomic groups through targeting of the 16S ribosomal RNA with oligonucleotide probes (21). It enables the tracing of taxonomically defined populations across environments and through time (21). In addition to abundance data, FISH allows conclusions to be drawn about microbial growth activity. Hybridized cells from highly active microbial populations appear, on average, larger compared to less active cells (22). Cell volumes may be derived from the FISH signal area, which is a two-dimensional representation of the cell volume (or more precisely the cytosol) (23). At the same time, FISH signal intensities correspond to cellular ribosome content, reflecting the potential for protein synthesis and thus growth. Previous studies have determined a linear correlation between ribosome contents and the growth rates for individual taxa (24–26). Prior to cell division, cells segregate their replicated genomes into the maturing daughter cells. Combining FISH with a DNA stain shows the intracellular DNA distribution, allowing the study of the frequency of dividing cells (FDC) (27, 28). The FDC has a linear correlation with the uptake of radio-labeled substrates (29) but has rarely been used in microbial ecology (30). Metagenomics has also been suggested for studying growth activities, as it would allow for a higher taxonomic resolution down to the species level. Most circular bacterial genomes are bidirectionally replicated. In short-read metagenomes, actively dividing cells are expected to have higher coverage of the origin of replication than their termini (31).

We sampled spring phytoplankton blooms at the long-term ecological research station (LTER) Helgoland Roads in the German Bight in 2018 and 2020. We used 16S rRNA-FISH and taxon-specific image cytometry to study cell volumes, ribosome content, and the FDC for *Bacteroidetes*, *Aurantivirga*, SAR86, and SAR11. Using taxon-specific cell division rates from dilution experiments, we calibrated FDC values to calculate cell division rates across a spring bloom. We also determined FISH-derived net growth rates and calculated mortality based on net growth and cell division rates. Our microscopy results were contextualized with data derived from the analyses of corresponding metagenomes.

## RESULTS

The 2018 and 2020 spring phytoplankton blooms at the LTER station Helgoland Roads were diatom dominated (9, 32), as in previous years (2). In both years, microbial cell counts increased after increases in the chlorophyll a concentration, which marked the onset of the spring phytoplankton blooms. In 2018, the chlorophyll a concentration increased from 0–2 µg/L (March till mid-April) to 6.7 µg/L on April 27. The total DAPI cell counts increased fourfold from $0.8 \times 10^6$ cells/mL (April 30) to $3.2 \times 10^6$ cells/mL on May 24 (Fig. S1A). In 2020, chlorophyll a concentration increased from below 1 µg/L at the end of March to 7 µg/L (April 26) and 9.4 µg/L (April 28). The total microbial cell counts increased approximately threefold from $0.6 \times 10^6$ cells/mL (mid-April) to $1.6 \times 10^6$ cells/mL (April 20), then collapsed to below pre-bloom conditions, and finally increased to $1.8 \times 10^6$ cells/mL on May 26 (Fig. S1B).

### Frequency of dividing cells as a robust parameter to investigate cell division

In 2020, SAR11 cell counts followed the general patterns of the total microbial counts (Fig. 1A; Fig. S1). Their abundance decreased toward the end of March from 1.9 to 0.7 $\times 10^5$ cells/mL (Fig. 1). Thereafter, cell counts increased and peaked on April 20 ($3.9 \times 10^5$ cells/mL), decreased until May 4 ($0.7 \times 10^5$ cells/mL), and increased again until the end of the sampling campaign. During the same period of time, the average cell volume, based on FISH signals, increased by a factor of ~1.5 and showed opposing trends to the cell counts (Fig. 1B). Average cell volumes increased from $0.10 \pm 0.05$ µm$^3$ (mean $\pm$ SD) on March 2 to $0.15 \pm 0.06$ µm$^3$ on April 1. Thereafter, the volumes decreased until April 20 ($0.11 \pm 0.04$ µm$^3$), when cell counts were maximal, but subsequently increased during the first week of May ($0.18 \pm 0.07$ µm$^3$ on May 4). The number of ribosomes per cell, determined using FISH fluorescence, showed a similar pattern to the cell volumes. They increased by a factor of ~2 from the beginning of March to April from $2.0 \pm 1.3$ to $4.1 \pm 2.2$ arbitrary units (AU), then decreased until April 20 ($2.1 \pm 1.1$ AU), and again increased in the first week of May ($5.0 \pm 2.5$ AU; Fig. 1C). The trends in the FDC concur with cell volume and ribosome content data (Fig. 1D). The FDC increased approximately threefold from around 4% in early March to a maximum of 12.6% on March 27, dropped to pre-bloom conditions until mid-April, and peaked a second time on May 8 (12.5%).

The three cellular characteristics (cell volume, ribosome content, and FDC) were not only positively correlated with each other for SAR11 (Fig. 2) but also for the three other taxonomic groups *Bacteroidetes*, *Aurantivirga*, and SAR86 (Fig. S2). The multiple linear regressions between the three characteristics among themselves, each with the additional interaction terms of the sampled year and the respective FISH probes, were statistically significant ($P < 0.0001$; further details are given in supplementary information). Due to the reasons discussed below, we proceeded with FDC as a suitable proxy of cell division rates, though all three characteristics would be suitable.

### Growth activity changes of SAR11, SAR86, *Bacteroidetes*, and *Aurantivirga* in 2018

We could only assess relative growth activity changes by studying FDC in 2018, as calibrations of FDC with dilution experiments were only done in 2020. The FDC values

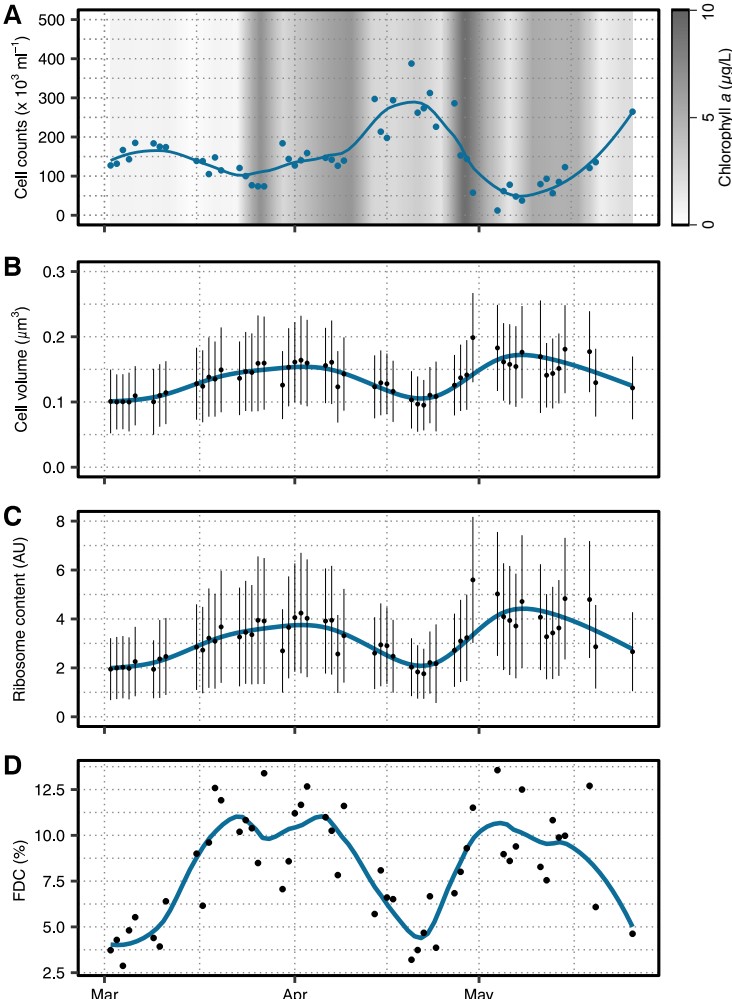

**FIG 1** Cellular parameters of SAR11 during the spring bloom in 2020. (A) Cell abundances in blue and chlorophyll a concentration as gray background. (B) Cell volumes and (C) ribosome contents were calculated from CARD-FISH signals and plotted as means per day (black points) ± SD (black lines). A loess smoothing of all data is depicted in blue. (D) The FDC, as a measure of cell division, was determined from cells with two intracellular local DAPI maxima. An FDC per sampling day is shown as black points and loess smoothing as blue line.

ranged between 5% and 15% with a few exceptions, mainly within the genus *Aurantivirga*. For SAR11, the FDC was initially between 8% and 10% from March 1 to April 11 but increased thereafter to ~15% by April 13. This increase occurred notably before chlorophyll a concentration started to increase by the end of April. The SAR11 FDC started to decrease after May 4 to pre-bloom conditions. SAR11 cell counts exceeded $2.5 \times 10^5$ cells/mL by April 3 and steadily increased to $1.1 \times 10^6$ cells/mL by May 24 (Fig. 3). SAR86 FDC increased from 4% to 8% until April 3, doubled to 16.5% by April 30, and decreased thereafter to around 10%. SAR86 cell counts were between 1 and $3 \times 10^4$ cells/mL until May 8, then abundances increased >10-fold, peaking at $3.5 \times 10^5$ cells/mL on May 24 (Fig. 3). *Bacteroidetes* FDC was initially low (4.7%–6.6%) until April 9, increased thereafter to reach 16.3% on April 26, and decreased afterward to pre-bloom conditions. *Bacteroidetes* cell counts varied between 0.6 and $1.7 \times 10^5$ cells/mL until May 3, peaked at $2.7 \times 10^5$ cells/mL on May 9, and peaked a second time with $6.2 \times 10^5$ cells/mL on May 24 (Fig. 3). *Aurantivirga* FDC was low in March (2%–6%; Fig. 3) and increased to a maximum of 20%

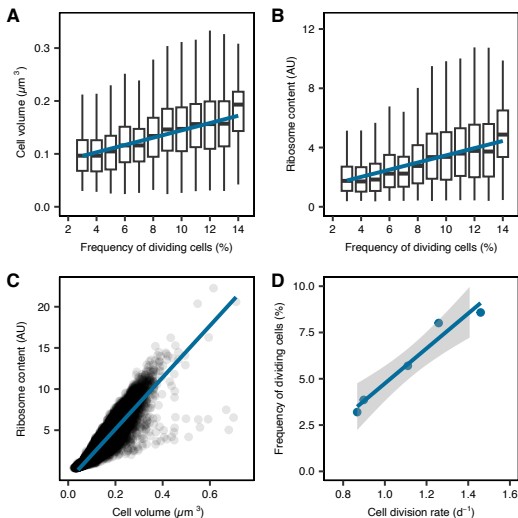

**FIG 2** Correlation of cellular parameters and cell division rates. (A) Box and whiskers plot of cell volume in relation to FDC with regression in blue. (B) Box and whiskers plot of ribosome content (fluorescence per cell) in relation to FDC with regression in blue. Boxes in (A) and (B) represent 25th and 75th percentile, and the mean is drawn as solid line within the box. The whiskers are 1.5× interquartile percentile. Outliers are not visualized. (C) Ribosome content plotted over measured cell volume as black points with linear regression is depicted in blue. (D) Cell division rates were assessed in dilution experiments and correlated with FDC. SE is shown as gray shading.

on April 27. In early May, FDC was around 2.5% but increased to 13% by May 28. *Aurantivirga* cell counts ranged between 0.1 and $0.9 \times 10^4$ cells/mL until mid-April when they started to increase to peak first on May 7 ($3.4 \times 10^3$ cells/mL) and again on May 24 ($9.3 \times 10^3$ cells/mL; Fig. 3).

## 2020 spring bloom cell division rates for SAR11, SAR86, *Bacteroidetes*, and *Aurantivirga*

We conducted dilution experiments on 5 days across the 2020 spring bloom to experimentally determine taxon-specific cell division rates (Table S1 at doi.org/10.6084/m9.figshare.22290166). We used multiple linear regressions with the null hypothesis that (i) FDC (FDC) is independent of experimentally derived cell division rates ($\mu$) and (ii) this relationship is independent of the assessed taxon (taxon; formula: FDC~$\mu$*taxon). We rejected both null hypotheses ($R^2 = 0.86$; $P < 0.0001$) and could calculate taxon-specific cell division rates from the FDC across the 2020 spring bloom (Fig. S3A; Table S2 at doi.org/10.6084/m9.figshare.22290166).

   Cell division rates varied noticeably over the course of the bloom of 2020. Generally, SAR11 and *Bacteroidetes* grew at rates of 0.5–2/day. SAR86, on the other hand, exceeded 0.5/day only once in late April. Please note that calculated cell division rates for SAR86 might be underestimated due to a single data point (Fig. S3A). However, they did not exceed rates of 0.6/day in the dilution experiments. *Aurantivirga* cell division rates were highly variable, ranging from no cell division to 1.9/day. In detail, SAR11 cell division rates increased ~threefold in March, even before the phytoplankton bloom started. Cell division rates reached their first maximum of 1.9/day on March 27, 1 week prior to the maximum in chlorophyll a concentration. It is remarkable that cell counts decreased to about half during the same time. Subsequently, cell division rates decreased to pre-bloom levels in mid-to-end of April (0.8/day and 1.2/day), when cell counts increased to reach a maximum. Furthermore, SAR11 cell division rates were >1/day on 43 of 53 sampling days in 2020 (Fig. 4). By contrast, SAR86 divided <0.5/day on 52 sampling days (Fig. 4). The average *Bacteroidetes* cell divided ~1/day in pre-phytoplankton bloom

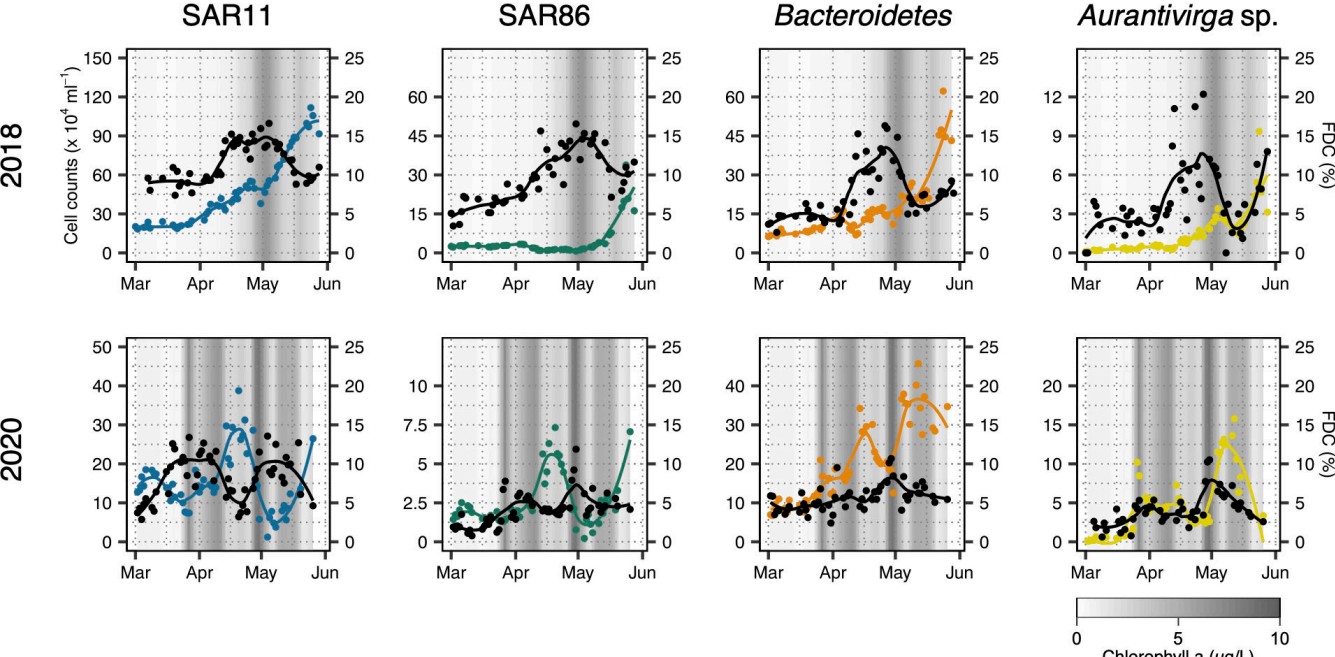

**FIG 3** Cell abundances and FDC in 2018 and 2020. Cell abundances of SAR11, SAR86, *Bacteroidetes*, and *Aurantivirga* during the spring bloom 2018 (upper row) and 2020 (lower row) in colored points with loess smoothing as colored lines. Taxon-specific FDCs are shown by black dots and loess smoothing as lines. Chlorophyll a concentration is shown as gray shading in the background in all plots.

conditions. Their cell division rate reached a maximum of 2.1/day on April 29, shortly before cell counts reached a maximum of $4.6 \times 10^5$ cells/mL (Fig. 4). *Aurantivirga* cell division rates covered the greatest range. While calculated rates were between 0 and 0.5/day pre-bloom, they peaked at 0.9/day on March 27 and 1.9/day on April 30, coinciding with an overall increased cell abundance *in situ* (Fig. 4).

## Cell division rates versus net growth rates during 2020 spring bloom

Besides cell division rates (calculated from the FDC), we also determined the net growth rate, based on the FISH abundance data. Net growth rates for all taxa ranged between −0.6 and 0.5/day, with two exceptions for *Aurantivirga* and one for SAR11 and SAR86, each (Fig. 4). These net growth rates were corresponding to doublings in cell abundances spanning multiple days. For example, the approximate doubling in SAR11 cell counts from $1.4 \times 10^5$ cells/mL (April 9) to $3.0 \times 10^5$ cells/mL (April 14) occurred within 5 days, which corresponds to a net growth rate of 0.15/day. The net growth (*r*) of, for example, SAR11 was almost an order of magnitude lower (minimum/maximum: −0.6 to 0.39/day, with one exception) than the cell division rates (*μ*, 0–2/day). It follows that the calculated mortality rates ($d = μ − r$) were high and close to the cell division rates (Fig. 4). We compared these calculated *mortality* rates to *grazing* rates, which were determined in the dilution experiments. Both were significantly correlated in a multiple regression model of *grazing ~ mortality\*taxon* ($R^2 = 0.86$; $P = 0.002$; Fig. S3B). The regression for SAR86 and *Bacteroidetes* was negative due to the spread of the data. Nevertheless, data from all taxa combined following the 1:1 ratio or calculated mortality were larger than grazing. This indicates that our calculated mortality rates can to a large extent be explained by grazing, with few cases where, for example, viral lysis might play an important role. Similarly, Sanchez et al. (33) found in a recent study that mortality due to grazers was larger than viral lysis, across multiple seasons and bacterial taxa.

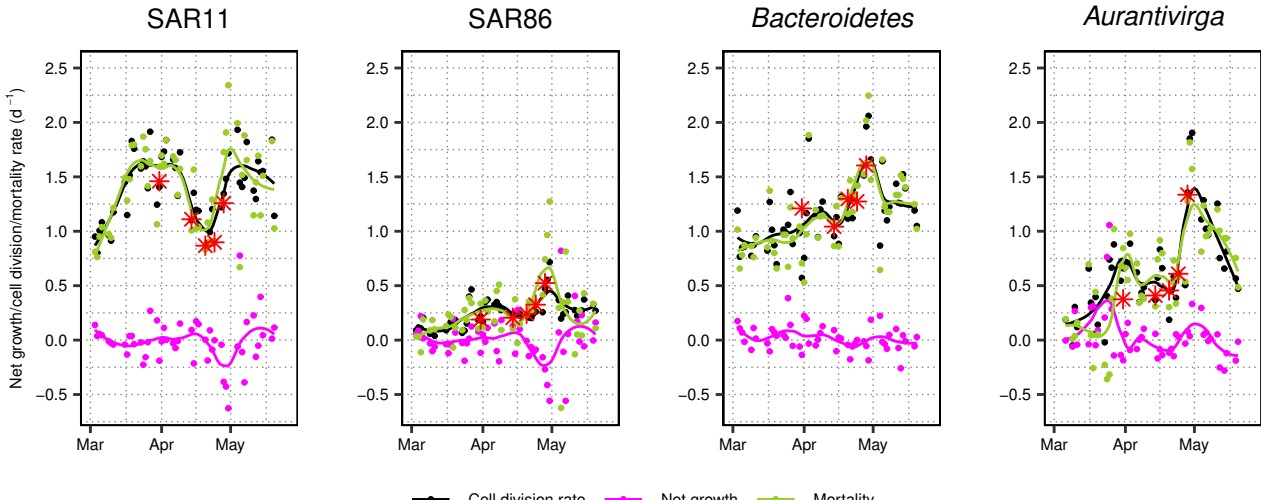

**FIG 4** Taxon-specific net growth, cell division, and mortality rates during 2020 spring bloom. Taxon-specific cell division rates were calculated based on FDC throughout the spring bloom (black points with loess smoothing as black line). Net growth was calculated based on FISH abundance data (magenta points with loess smoothing as magenta line). Mortality is the cell division rate minus net growth (green points with loess smoothing as green line). Measured cell division rates by dilution experiments are depicted with a red asterisk.

## Bioinformatic assessment of taxon diversity and growth measures during the 2018 bloom

Two metagenomes were sequenced per week during the 2018 sampling campaign. We compared our cytometric results to metrics derived from these metagenomes. First, we assessed the diversity of retrieved representative metagenome-assembled genomes (MAGs) from the four studied taxa across the spring bloom. We checked whether community shifts within one taxon could be responsible for the observed changes in abundances, FDC, and growth rates. SAR11 was represented by five MAGs, of which four belong to the open ocean clade 1a.1 and MAG r31 to clade 3 (Fig. S4) (34, 35). Both SAR11 and SAR86 were dominated by a single MAG toward the end of the bloom (Fig. S5). For *Bacteroidetes*, different species of the family *Flavobacteriaceae* succeeded each other. First, MAGs belonging to the GTDB-Tk genus-level clade MAG-121220-bin8 were most abundant until mid-April and were followed by the genus-level clade Hel1-33-131 (Fig. S5; Table S3 at doi.org/10.6084/m9.figshare.22290166). Within the genus *Aurantivirga*, MAG r29 initially dominated until mid-April and then MAG r261 took over until the end of April (Fig. S5).

Next, we aimed to assess microbial growth parameters during the 2018 spring bloom using *g*rowth *r*ate *ind*ex (GRiD) values. We tested different mapping algorithms, which resulted in substantially different GRiD values, while the estimates of sequencing depth for individual MAGs were comparable between methods (Fig. S6). Here we focus on GRiD values obtained using default settings. GRiD values fluctuated between 1.1 and 2.8 for all the assessed MAGs. However, no SAR86 MAG exceeded a GRiD value of 2, in contrast to the three other groups. GRiD values of the most abundant MAGs exhibited little variability over the spring bloom (Fig. S5). For example, SAR11 MAG r27 was the most abundant, especially toward the end of the spring bloom, but had low GRiD values compared to other SAR11 MAGs (Fig. S5). The determined GRiD values correlated positively to FDC, with a taxon-specific interaction term, though with high variance ($p <$ 0.0001, $R^2 = 0.12$; Fig. S5).

Finally, we used the codon usage bias method gRodon to predict the possible maximum cell division rates (Fig. 5). Four out of five assessed SAR11 MAGs were identified as oligotrophs. SAR11 MAG r116 had a predicted maximum growth rate of 10/day (Table S4 at doi.org/10.6084/m9.figshare.22290166) but had among the

lowest relative abundances (≤1% and absent in mid to end May) throughout the 2018 phytoplankton bloom. All SAR86 MAGs were identified as oligotrophs and the *Aurantivirga* MAGs as copiotrophs. The phylum of *Bacteroidetes* was rather heterogeneous, with 48 MAGs classified as copiotrophs and 38 MAGs as oligotrophs (Fig. 5 and Table S4 at doi.org/10.6084/m9.figshare.22290166). Interestingly, the *Bacteroidetes* GTDB-tk genus MAG-121220-bin 8, the first dominating *Bacteroidetes* genus, was classified as an oligotroph (maximum growth rate <4/day), while Hel1-33-131, the most abundant MAG toward the end of the sampling, is classified as a copiotroph (maximum growth rate 6.9/day).

## DISCUSSION

We studied taxon-specific growth changes using *in situ* image cytometry during the course of two spring diatom blooms. FDC is the method of choice to assess cell division activity, though all FISH-derived parameters, namely cell volume, ribosome content, and FDC, co-varied over time (Fig. 1; Fig. S2). First, FDC quantifies the proportion of actively dividing cells, whereas cell volume and ribosome content are indirect measures of cellular growth spreading over a continuum of values with no defined threshold of cell division. Second, FDC is methodologically advantageous over the former two, as it combines two separate stains. Hence, the object identification (i.e., FISH-positive cell) and the measured property (i.e., DAPI distribution) are effectively independent of each other (36). Next, relative differences were most pronounced for FDC, allowing the detection also of small changes in microbial growth. Furthermore, FDC correlated linearly with taxon-specific cell division rates determined in dilution experiments (Fig. S3), corroborating earlier findings from pure cultures and environmental samples (37, 38).

Our image cytometry approach had some limitations regarding cell volume measurements and dilution experiments. First, cells are filtered onto polycarbonate filter and might lose some of their height due to fixation. Therefore, our three-dimensional models of cell volumes most likely somewhat overestimate in the third dimension. Second, cell volume measurements are derived from a CARD amplification signal, which often seems to overshadow the cell boundaries and hence overestimate the cell dimensions. Additionally, object identification and volume measurement were both done on the same signal. The thresholds to identify a cell immediately influence the cell size and volume estimates (36). Taken together, this could contribute to an overestimation of cell volume measurements. Nevertheless, this should not affect comparisons of cell volumes within this study. Finally, our dilution experiments did not exclude phage-free cell division rates, as other studies have done (33). The dilutions were prepared with 0.2-µm filtered water, which is larger than most phages.

We challenged the FDC-derived cell division rates using bioinformatic predictions from metagenomes and MAGs for the 2018 spring bloom. We computed GRiD (31), which was highly susceptible to the mapping tools that were used, not yielding any reproducible results. Therefore, we cannot support using the GRiD algorithm at this developmental stage. GRiD values generated under default mode were generally correlated with the taxon-specific FDC, though with little predictive power ($R^2 = 0.12$). Although GRiD could in theory be useful to assess individual species or strains to a higher taxonomic resolution than FISH-based microscopy, microscopically derived growth measures remain more direct and precise. In addition, we computed gRodon values to predict the genomic potential for maximum cell division rates. They can be used to categorize the retrieved MAGs as copiotrophic and oligotrophic, according to the authors of gRodon (39). The gRodon results were in line with our assumptions that SAR11 and SAR86 can be considered oligotrophs and *Aurantivirga* a copiotroph. *Bacteroidetes* being heterogeneous, with the majority of clades putatively slow growing, confirms previous findings of few actively growing *Bacteroidetes* clades during phytoplankton blooms (5). All experimental cell division rates were slower than gRodon-predicted genomic potentials for maximum cell division rates, which indicates that—on a community level—none of the assessed groups divides to their full capacity.

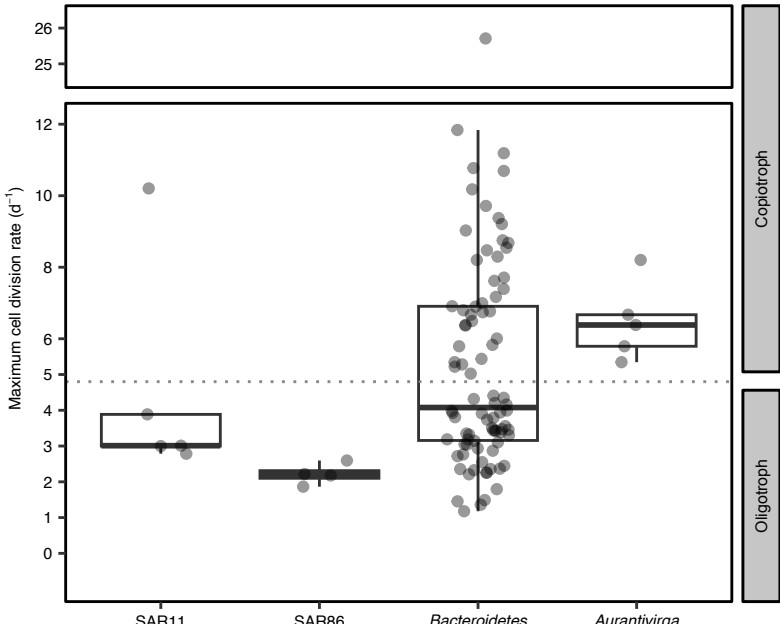

**FIG 5** gRodon-predicted minimal doubling times. Box and whisker plots of genomic potential for minimal doubling times predicted by gRodon for SAR11, SAR86, *Bacteroidetes*, and *Aurantivirga* MAGs, retrieved from 2018 spring phytoplankton bloom. Boxes indicate 25th and 75th percentile, and the mean is drawn as solid line. Whiskers represent 1.5× interquartile range, and outliers are not visualized. Points indicate the results of individual MAGs. Dotted line indicates threshold between oligotrophs (minimal doubling time >5 hours) and copiotrophs (minimal doubling time <5 hours), according to the authors of gRodon.

Under constant substrate and nutrient conditions, cell division and mortality rates are both temperature dependent (40, 41). This is also known for bacteria in environmental samples (e.g., 42, 43) but only partly visible in our case (Fig. S1 and Table S5 at doi.org/10.6084/m9.figshare.22290166). For example, though the temperature increased between April and May 2020 from 9.9°C to 11.4°C (Fig. S1B), the cell division rates of *Bacteroidetes* and *Aurantivirga* decreased in May, and SAR11 cell division rates fell to pre-bloom levels in mid-April and end of May. Other than temperature, the bacterial communities are shaped by phytoplankton-derived organic matter (1, 2). Inorganic nutrients such as nitrate, ammonium, phosphate, and silicate, which are tightly monitored at the LTER Helgoland, are negatively correlated with FDC (Table S5 at doi.org/10.6084/m9.figshare.22290166) (2020 data: 9 and 2018 data: 32). However, these nutrients are directly taken up and depleted by phytoplankton and are, thus, only indirectly correlated with FDC without causation (2, 32).

The diverse phylum *Bacteroidetes* comprised fast and slow-growing bacteria. The observed cell division rates for *Bacteroidetes* (minimum to maximum: 0.6 to 2.1/day) agree with previous reports, ranging from 0.5 to 5.1/day (6, 7, 20, 44). Metagenome analyses confirmed that the *Bacteroidetes* constituted a highly diverse phylum with a large variety in minimal doubling times, predicted from MAGs using gRodon, and large variations in growth as reconstructed by GRiD. Thus, the division rates of individual *Bacteroidetes* species might be considerably higher than those for the remainder of the community.

The genus *Aurantivirga* is known as one of the first responders to phytoplankton blooms, not only in the North Sea (e.g., 45) but also in polar waters (46, 47). In this study, *Aurantivirga* showed the greatest plasticity in cell volume and cell division rates (0 to 1.9/day) over the course of the 2020 spring bloom, with a pronounced peak during the

later bloom stages. *Aurantivirga* has previously been found to outcompete other taxa by their capability to digest algae-derived polysaccharides (5, 9, 45). This fits with the general observation that *Aurantivirga* net growth and cell division rates increased with the peaks in chlorophyll a in both years, although the net growth and cell division rates were statistically not correlated. Our metagenome-derived gRodon results indicate that all *Aurantivirga* MAGs have minimal doubling times typical for copiotrophs. We conclude that the *Aurantivirga* populations had a copiotrophic lifestyle with rapid boom and bust cycles.

SAR86 cell division rates increased at the beginning of the 2020 bloom, which indicates the dependence of SAR86 on organic matter exudated by live phytoplankton (48). At the end of April, SAR86 cells were apparently exposed to changes in top–down control factors, as its net growth bottomed (−0.1 to −0.2/day) while the cell division rates peaked (>0.5/day). SAR86 cell volumes were comparable (0.16 to 0.40 $\mu m^3$) to *Bacteroidetes* (0.22 to 0.53 $\mu m^3$) and therefore much larger than previously reported (0.06 to 0.08 $\mu m^3$) (22), but this might be an overestimation (please see the critical evaluation of our cell volume measures above). SAR86 was the only taxonomic group for which the MAGs never exceeded GRiD values of 2 throughout the spring bloom. Likewise, the gRodon values characterized all SAR86 MAGs as oligotrophic. SAR86 cell division was among the lowest in our study, not only from the dilution experiments (max. 0.6/day) but also from the calculated rates throughout the spring bloom (<0.75/day).

SAR11 comprised the smallest cells of all four groups in our study with the least variability in cell volume. Due to the assumed slower cell division rates of SAR11, we hypothesized less variation in ribosomal content compared to putatively faster growing *Bacteroidetes* and *Aurantivirga* (49). While generally lower, the ribosomal content of SAR11 cells fluctuated comparable to the three other taxa. They divided faster (max. 1.9/day) than cultivated SAR11 in optimized media (<0.5/day) (13, 14, 50). However, our findings are in line with previous SAR11 cell division rates from dilution experiments (1.2 to 1.8/day) from coastal Mediterranean waters (33, 51, 52). The here-assessed coastal SAR11 cells were dominated by members of the clade 1a.1, which is commonly attributed to the open ocean, in 2018 (Fig. S5) and 2020 (9).

To our surprise, SAR11 increased their cell division rates days to weeks before the main phytoplankton bloom started in both studied years. SAR11 cells, like SAR86, *Aurantivirga*, and other *Bacteroidetes*, are potential photoheterotrophs capable of proteorhodopsin-dependent ATP synthesis (16, 18, 53). Hence, increasing light intensities during the spring blooms could support growth by fueling energy-dependent transport albeit only SAR11 cells seemed to have benefitted from this. Above ~25 Einstein/$m^2$/day, SAR11 cell division was increased gradually (Fig. S1 and S7), which could potentially be considered as a threshold in our case to obtain enough energy for increased activity. Previous incubations detected increased proteorhodopsin-derived activity in SAR11 after incubations with 36 Einstein/$m^2$/day (54).

Despite high cell division rates (>1/day) before the phytoplankton bloom 2020, SAR11 cell abundances did not increase and even decreased. Net growth was almost an order of magnitude lower than the cell division rate, indicating tight top–down controls. Since SAR11 exhibited the highest mortality rates before the phytoplankton bloom, we assume a SAR11-specific top–down control factor. Non-specific grazing is rather unlikely due to the small cell size (55), and taxon-specific grazing (56, 57) has to yet be shown for SAR11. Besides grazing, viruses are known to shape the SAR11 community (51, 58, 59). Previous dilution experiments in the Mediterranean Sea accounted for viral lysis. The authors found an increased influence of viruses on the SAR11 community, especially during autumn but less in spring (33). Considering the overall dynamics in our data, the timing of sampling is crucial and could explain why similar earlier experiments did not detect an effect of viruses on the SAR11 community (51).

All in all, it is not only their high cellular abundance (48) but also mainly the high cell division and mortality rates that impact our perception of the microbial loop and thereby the entire marine carbon cycle. Based on our data, we propose that the clade SAR11 not

only consists of clear-cut oligotrophs but that coastal strains (including clade 1a.1) also exhibit fast cell division rates typical of copiotrophs.

## Constant struggle of microbes against mortality

In addition to cell division rates, we also estimated net growth rates based on FISH abundance data. The difference between these two rates yields corresponding mortality rates. For all studied taxonomic groups, the cell division and mortality rates were generally close to each other. We validated our derived mortality rates against the grazing rates obtained from the dilution experiments, which largely followed a 1:1 ratio. A 1:1 ratio would mean that all mortality is due to grazing. Our results reveal that despite several cell divisions per day, mortality diminishes the increase in cell abundance. In other words, cell division rates in the environment are higher than anticipated, however, mortality removes >90% of newly produced bacterial biomass each day. Similar tight couplings have been reported earlier (33), though with great variation throughout an entire year but not resolved on a temporal scale as our data. In this context, it is also interesting to note that mortality sets in almost instantaneously with no detectable delay. This means that grazers and/or phages are present and ready to control the growing community effectively at all times. This scenario also seems to be the rule rather than the exception, as we could demonstrate such a tight trophic coupling in four taxonomic groups for two spring blooms with high primary production covering a period of 3 months each.

## Conclusion

In our study, we have shown that FDC values enable the measurement of cell division rates *in situ*, after taxon-specific calibrations. This is rather straightforward to implement in future studies. Based on the changes in FDC over time, we showed evidence of an interplay between bottom–up and top–down controls in the early phase of spring phytoplankton blooms. These results raise many questions. For example, what fuels SAR11 to grow at high cell division rates of 1.9/day before the phytoplankton bloom? Similarly, what are the top–down controls that balance these fast division rates? In the past, much research has focused on bottom–up control factors. With the tools presented here, future research may include top–down control factors, which equally shape the bacterial world.

## MATERIALS AND METHODS

### Sampling

Marine surface water (~1 m depth) was sampled at the LTER Helgoland Roads (54° 11.3' N, 7° 54.0' E) (60) in a well-mixed pelagic water column during spring phytoplankton blooms in 2018 and 2020. Microscopy samples were collected every working day between March and June by the research vessel Aade. We fixed samples of 10 mL (SAR11 and *Bacteroidetes*) or 100 mL (SAR86 and *Aurantivirga*) with 0.2-µm-filtered formaldehyde (1% final concentration, 1 hour at room temperature). These different volumes were necessary to account for variation in abundance. Subsequently, fixed cells were filtered onto 0.2-µm polycarbonate filters (47 mm diameter; Sigma Aldrich, Taufkirchen, Germany) and placed on 0.45-µm cellulose nitrate support filters (Sigma Aldrich). The filters were stored at −20°C until further processing. Chlorophyll a concentration was measured twice a week via high-performance liquid chromatography using the method of Zapata et al. (61) and Wiltshire et al. (62). Photosynthetically active radiation (PAR) remote sensing data were retrieved for the sampling period from the NASA Goddard Space Flight Center (63). Data were analyzed with the R package raster (64) cropped to cover the German Bight (53° 41' 17.8794" to 54° 41' 17.8794" N and 7° 24.0' to 8° 24.0' E). A loess average of the cropped data was visualized (Fig. S1).

For metagenomic sequencing, seawater was sampled at 1 m depth twice a week over a period of 3 months. One liter of unfixed seawater was sequentially filtered through 10, 3, and 0.2-µm pore-sized polycarbonate filters. Filters were flash-frozen in liquid nitrogen and stored at −80°C until further use.

## Cell division rates based on dilution grazing experiments

We conducted five dilution grazing experiments before, during, and after the phytoplankton bloom of 2020 (March 31 and April 14, 20, 24, and 28) to determine the cell division rates of individual bacterial clades. Seawater was sampled at Helgoland Roads and sieved (200 µm) to exclude mesozooplankton such that the only consumers were microzooplankton and heterotrophic nanoflagellates. This water was subsequently diluted with 0.2-µm sterile-filtered seawater to create a dilution series of 100% (undiluted), 50% (1:1), 25% (1:3), and 10% (1:9) in 1-L cell culture flasks (Greiner, Kremsmünster, Austria). No further nutrients were added. One aliquot of the undiluted samples was taken as a reference at the beginning of sampling ($t_0$). All dilutions and 24-hour incubations were prepared in duplicate.

The flasks were placed on a plankton wheel (~3.2 rpm) to prevent the sedimentation of the planktonic organisms and incubated with a day-to-night regime of 14-10 hours (20 to 30 µmol photons/m$^2$/s) for 24 hours in a temperature-controlled room set at the *in situ* sea surface temperature of the corresponding day. After 24 hours, samples were taken from all the duplicated dilutions, and fixed and filtered as described for the microscopy samples.

Total and taxon-specific cell concentrations were determined through DAPI staining and FISH experiments similar to those for the environmental samples, as described below and in detail in the supplementary information. Cellular concentrations from the dilution experiments are provided in Table S1 (at doi.org/10.6084/m9.figshare.22290166). Cell division rates were calculated following Landry and Hassett (65), as described in the supplementary information.

## Cell counts and FISH

Samples were stained with the DNA stain 4′,6-diamidino-2-phenylindole (DAPI; 1 µg/mL, 7 minutes at room temperature) and subsequently washed with deionized water and ethanol. Catalyzed reporter deposition (CARD)-FISH was performed with probes targeting SAR11 (SAR11 mix), SAR86, *Bacteroidetes* (CF319a), and *Aurantivirga* (AUR452; Table S6 at doi.org/10.6084/m9.figshare.22290166) following the protocol described in Fuchs et al. (66) and in more detail in the supplementary information. The non-sense-probe NON338 was included as a negative control. All probes were purchased from Biomers (Ulm, Germany).

We excluded a potential impact of the CARD signal amplification on the linearity of the fluorescence measurements and cell volume determinations by a comparison between CARD-FISH and tetra-labeled FISH on selected samples on the 2020 dataset (Fig. S8). All samples were embedded in antifading media Citifluor:Vectashield (1:3; Citifluor, London, UK; Vector Laboratories, Burlingame, CA, USA) for microscopy.

## Automated image recording

Images were recorded on a Zeiss AxioImager.Z2m microscope with a cooled charged-coupled device (CCD) camera (Zeiss AxioCam MRm, Zeiss Oberkochen, Germany). The microscope was equipped with a Zeiss Colibri 7 LED (385 nm for DAPI, 469 nm for Alexa 488 dye, and 590 nm for autofluorescence) and a Multi Zeiss 62 HE filter cube (Beam splitter FT 395 + 495 + 610). The Zeiss AxioVision software (Zeiss, Germany) was used for automated image acquisition with a custom-built macro (23, 67, 68). The focal planes of 120 fields of view per sample were identified with 1× magnification. Subsequent fine-tuning and image recording were done with a 63x Plan Apochromat objective (1.4 numerical aperture, oil immersion).

## Image cytometry

The obtained 8-bit grayscale images were loaded into our Automated Cell Measuring and Enumeration tool (ACME, available from https://www.mpi-bremen.de/automated-microscopy.html) for manual curation and image analysis (Fig. S9) as described previously (67, 68). FISH signal–derived cell size and signal intensity measurements were exported from the ACME tool. To calculate cell volumes $V$ from the FISH signal–derived cell sizes (two-dimensional projection), we used the basic geometric approximation of cylinders with hemispherical caps (69–71), with the cylinder radius $r$ and length $l$ (i.e., total length $l_{tot} - 2r$): $V = \frac{4}{3}\pi r^3 + \pi r^2 l$. Previous research has shown differences as low as ~1% to more sophisticated models (23).

We measured the total fluorescence (i.e., the sum of gray values of all pixels within one cell), based on the FISH signal. Additionally, the image processing software imageJ/Fiji (v2.1.0/1.53e, 72) with the plug-in MicrobeJ (v5.13l, 73) was used to calculate the FDC. A FISH-positive cell was defined as dividing if it contained two local DAPI maxima (compared to one local maximum for non-dividing cells). FDC was calculated for each taxon individually as FDC = Σ(dividing cells)/Σ(all cells). A more thorough description for the ACME tool and MicrobeJ image processing are provided in the supplementary information.

## DNA extraction, metagenome sequencing, and diversity and growth estimation

DNA from free-living bacteria from the 0.2–3 µm fraction was extracted following Zhou et al. (74) and quantified on a NanoDrop 2000c spectrophotometer (Thermo Scientific, Waltham, MA, USA). The DNA concentrations ranged from 3 to 45 ng DNA/µL. Extracted DNA was sequenced at the Max Planck Genome Centre, Cologne. The sequencing was performed with PCR-free DNA library type on an Illumina HiSeq 2500 platform (rapid mode) with 2 × 250 base pair chemistry (San Diego, CA, USA). Raw reads (accession numbers in Table S7 at doi.org/10.6084/m9.figshare.22290166) were quality trimmed and filtered using the *bbduk.sh* script of the BBMap suite (v35.14, 75) and assembled into contigs using SPAdes (v3.11.1, 76). Contigs were further binned within anvi'o (v6.2, 77) using sequencing depth from at least three other samples. Retrieved bins were manually refined by invoking the anvi-refine command within anvi'o. The quality of bin in terms of completeness and contamination was assessed by checkM (v1.0.18, 78). In total, 1,222 MAGs were retrieved, 852 of which were >50% complete and had <5% contamination (79). As assembly and binning was performed on individual samples, redundant MAGs were obtained. Dereplication of MAGs was performed using dRep (v3.0.0, 80) applying an average nucleotide identity of 99%. Taxonomic classification of representative MAGs was performed with GTDB-tk (v1.7.0) using GTDB r202 (81). MAGs belonging to SAR11 (g_*Pelagibacter*, $n = 5$), SAR86 (o_SAR86, $n = 4$), *Aurantivirga* (g_SCGC-AAA160-P02, $n = 10$), and *Bacteroidetes* (p_*Bacteroidota*, $n = 86$) were chosen based on their phylogenetic assignments. SAR11 MAGs were included in a reference tree for more detailed phylogenetic identification (see the supplementary information). MAG abundances were calculated as described in the supplementary information. MAGs were renamed with a consecutive number for this study. The original names, checkM quality scores, and gRodon results (see below) are described in Table S4 (at doi.org/10.6084/m9.figshare.22290166).

We determined the GRiD (31) for all MAGs during the spring bloom. We compared GRiD results using different settings for SAR11, SAR86, and *Aurantivirga*: default settings, default settings with reassignment of ambiguous reads, and our own mappings using bowtie2 (see the supplementary information). The main figures of this manuscript show GRiD values obtained from default settings.

Maximum growth rate estimations were calculated using the R package gRodon (39), which is based on codon usage bias. Taxa with higher growth rates are adapted to use DNA codons with the highest abundance of corresponding tRNAs in their cells

(82, 83). The authors of gRodon identified a threshold of 5 hours of minimal doubling time (≙ growth rate < 4.8/day). Below 5 hours of predicted minimal doubling times, the respective microbe is considered as copiotroph, while microbes with doubling times above this threshold are classified as oligotroph. We followed all the suggestions in the gRodon manual under default settings, including prokka genome annotation (84) and Biostrings R package usage (85).

## Determining growth, modeling, and statistical analyses

All modeling and statistical analyses were executed in R (v1.2.5042, 86). Calculated cell volumes, cellular fluorescence intensities, and FDC were modeled with local estimated scatterplot smoothing (loess [span = 0.4]).

We statistically tested the relationship of the modeled *FDC* over the experimentally derived cell division rates **μ** with an interaction term of the used *FISH probe* (*FDC ~ μ*FISH probe*). The estimated regression model, including the interaction term, was significant ($P < 0.0001$, $4.3 \times 10^{-5}$, $R^2 = 0.85$; see the supplementary information for the results of *post hoc* test). We proceeded with the model and used its coefficients to calculate cell division rates $\mu$, based on the FDC (Table S2 at doi.org/10.6084/m9.figshare.22290166).

We calculated net growth ($r$) using FISH-derived abundance data with a sliding window of five timepoints from: $r = (ln(N_{End}/N_{Start}))/t_{End} - t_{Start}$ with $N_{Start}$ and $N_{End}$, respectively, being the modeled abundance at timepoint $t_{Start}$ and $t_{End}$. For each timepoint, two preceding and two succeeding data points were included, as part of the sliding window. In cases where the linear regression resulted in negative values, net growth could not be calculated, as the natural logarithm is only defined for $x > 0$. Using net growth $r$ and cell division rate $\mu$, we also calculated mortality or death rates $d$ from: $r = \mu - d$.

## Visualizations

Data were organized and visualized using the R packages ggplot2 (v3.3.3, 87), plyr (v1.8.6, 88), lubridate (v1.7.10, 89), reshape2 (v1.4.4, 90), cowplot (v1.1.1, 91), ggpubr (v0.4.0, 92), gghalves (v0.1.3, 93), emmeans (v1.7.5, 94), and car (v3.1.0, 95). The color schemes were inspired by the Wes Anderson palette (v0.3.6, 96). All R scripts were uploaded to GitLab and are freely available (https://gitlab.mpi-bremen.de/jbruewer/bacterial-activity-manuscript-figures).

### ACKNOWLEDGMENTS

We would like to thank the captain and crew of RV Aade, Lilly Franzmeyer, Karl-Peter Rücknagel, Fengqing Wang, and Mikkel Schultz Johansen for collecting and processing samples during the 2018 and 2020 spring bloom on Helgoland. Karl-Peter Rücknagel, Jörg W. Wulf, and Kathrin Büttner assisted with CARD-FISH experiments and automated microscopy for the dilution experiments. We would like to thank Lisa Bauer for the CARD-FISH versus Tetra-FISH experiments. We are grateful to Nikolaus Leisch, Jakob Pernthaler, and Benedikt Geier for their valuable suggestions regarding data analysis. We thank Ben Francis and Fengqing Wang for MAG sequence submission and Hanno Teeling for critical reading and suggestions. Victor Kelly is acknowledged for language editing. Funding was provided by the German Research Foundation (DFG) project FOR 2406/2 "Proteogenomics of Marine Polysaccharide Utilization (POMPU)" by grants of B.M.F. (FU 627/2-2) and R.A. (AM 73/9-2) and by the Max Planck Society to J.D.B., L.H.O., C.S., R.A., and B.M.F.

The authors declare no competing interests.

### AUTHOR AFFILIATIONS

[1]Max Planck Institute for Marine Microbiology, Bremen, Germany
[2]Alfred-Wegener-Institut, Helmholtz-Zentrum für Polar- und Meeresforschung, Biologische Anstalt Helgoland, Helgoland, Germany

[3]University of Bremen, Bremen, Germany

[4]Alfred-Wegener-Institut, Helmholtz-Zentrum für Polar- und Meeresforschung, Wattenmeerstation, List auf Sylt, Bremerhaven, Germany

## AUTHOR ORCIDs

Jan D. Brüwer  http://orcid.org/0000-0002-7317-4613
Luis H. Orellana  http://orcid.org/0000-0002-2766-5243
Chandni Sidhu  http://orcid.org/0000-0003-1386-1881
Helena C. L. Klip  http://orcid.org/0000-0002-0699-3934
Cédric L. Meunier  http://orcid.org/0000-0002-4070-4286
Maarten Boersma  http://orcid.org/0000-0003-1010-026X
Karen H. Wiltshire  http://orcid.org/0000-0002-7148-0529
Rudolf Amann  http://orcid.org/0000-0002-0846-7372
Bernhard M. Fuchs  http://orcid.org//0000-0001-9828-1290

## FUNDING

| Funder | Grant(s) | Author(s) |
| --- | --- | --- |
| Max Planck Society | | Jan D. Brüwer |
| | | Chandni Sidhu |
| | | Rudolf Amann |
| | | Bernhard M. Fuchs |
| | | Luis H. Orellana |
| Deutsche Forschungsgemeinschaft (DFG) | FU 627/2-2 | Bernhard M. Fuchs |
| Deutsche Forschungsgemeinschaft (DFG) | AM 73/9-2 | Rudolf Amann |

## AUTHOR CONTRIBUTIONS

Jan D. Brüwer, Conceptualization, Data curation, Formal analysis, Investigation, Methodology, Project administration, Resources, Software, Visualization, Writing – original draft, Writing – review and editing | Luis H. Orellana, Formal analysis, Methodology, Writing – review and editing | Chandni Sidhu, Formal analysis, Writing – review and editing | Helena C. L. Klip, Formal analysis, Investigation, Writing – review and editing | Cédric L. Meunier, Formal analysis, Investigation, Methodology, Writing – review and editing | Maarten Boersma, Methodology, Writing – review and editing | Karen H. Wiltshire, Investigation, Writing – review and editing | Rudolf Amann, Conceptualization, Data curation, Formal analysis, Funding acquisition, Resources, Supervision, Validation, Writing – review and editing | Bernhard M. Fuchs, Conceptualization, Data curation, Funding acquisition, Methodology, Resources, Supervision, Visualization, Writing – review and editing

## DATA AVAILABILITY STATEMENT

Sequencing data are accessible via BioProject PRJEB38290. Scripts for visual representations and raw data retrieved from microscopy analysis are available on GitLab: https://gitlab.mpi-bremen.de/jbruewer/bacterial-activity-manuscript-figures.

## ADDITIONAL FILES

The following material is available online.

## Supplemental Material

**FIG S1 (297276_1_supp_6694947_rrs7bx.pdf).** Total DAPI-stained cell counts with chlorophyll a concentration (grey, background), temperature, and photosynthetically active radiation (PAR) during the spring phytoplankton bloom (A) 2018 and (B) 2020.

**FIG S2 (297276_1_supp_6694948_rrs7zx.pdf).** Taxon and year-specific correlation of cellular parameters measured with FISH. (A) Ribosome content to cell volume. (B) Ribosome content over FDC. (C) Cell volume over FDC. Box-whisker plots in (B) and (C) range from 25th to 75th percentile and the whiskers represent 1.5* interquartile range. Outliers are visualized by dots. Mean is drawn as a solid line inside the boxes. Statistic results of regressions are reported supplementary results.

**FIG S3 (297276_1_supp_6694949_rrs7sx.pdf).** (A) Linear correlations of microscopically-derived FDC and cell division rates determined by dilution experiments. Information about the linear regression can be found in table S2 (at doi.org/10.6084/m9.figshare.22290166). (B) Taxon-specific grazing rates over mortality rates in 2020. Taxon-specific grazing rates were determined with dilution experiments on 5 time-points during the 2020 phytoplankton bloom. Mortality rates were calculated from cell division rates and on net growth. Net growth rates could not be retrieved for SAR11 and SAR86 on one sampling day, as local regressions of abundance values were computed to calculate net growth and the regressions were partly negative. Black line is an ideal line of 1:1 correlation of grazing and mortality rates.

**FIG S4 (297276_1_supp_6694950_rrs7qy.pdf).** Phylogenetic tree with SAR11 MAGs that have >50% completeness and <5% contamination described in this study with previously published SAR11 single amplified genomes (SAGs) (Haro‐Moreno JM, Rodriguez‐Valera F, Rosselli R, Martinez‐Hernandez F, Roda‐Garcia JJ, Gomez ML, Fornas O, Martinez‐Garcia M, López‐Pérez M. 2020. Environmental Microbiology 22:1748-1763) and SAR11 isolates (Delmont TO, Kiefl E, Kilinc O, Esen OC, Uysal I, Rappe MS, Giovannoni S, Eren AM. 2019. eLife 8:e46497). Colours according to clade assignment (indicated in outer ring) according to the literature (see above).

**FIG S5 (297276_1_supp_6694951_rrs7sy.pdf).** Metagenome-based assessment of microbial growth in 2018. Relative abundances of metagenome assembled genomes (MAGs; A-D), as well as GRiD values (E–H), of all four taxa were calculated across the spring bloom 2018. For SAR11, SAR86, and *Aurantivirga* results of individual MAGs are visualized. For *Bacteroidetes*, results are summarized on the genus level (C, G with standard deviation). (I-L) MAG-derived GRiD values plotted versus FDC for phytoplankton spring bloom 2018.

**FIG S6 (297276_1_supp_6694952_rrs77y.pdf).** Comparison of GRiD values and abundance estimates from the 2018 phytoplankton spring bloom with different mappings for (A) SAR11, (B) SAR86, and (C) *Aurantivirga*. From left to right: GRiD values from customized alignment, retrieved with BBmap; GRiD software in default mode (minimum coverage: 5); GRiD software in default mode (minimum coverage: 5) and re-alignment of ambiguous reads with Pathoscope2 within the GRiD software.

**FIG S7 (297276_1_supp_6694953_rrs7gx.pdf).** Relationship of SAR11 FDC to PAR for 2018 and 2020. Left: FDC over Photosynthetically active radiation (PAR) on the left. A *loess* moving average is plotted. Right: PAR (ochre) and FDC (black) are plotted over the spring blooms with chlorophyll a plotted in the background. Scale is the same as Fig. 1 and 2. Red dashed line indicate potential threshold of 25 Einstein $m^{-2}$ $d^{-1}$ for SAR11 activity in the beginning of the bloom.

**FIG S8 (297276_1_supp_6694954_rrs7ry.pdf).** Correlation of CARD-FISH and tetra-labelled FISH cell volumes and signal intensities for *Bacteroidetes* and SAR11 cells from selected 2020 spring bloom dates. Displayed are means of each sampling day and linear regression of the means. The displayed statistics are for the linear regression model on the means.

**FIG S9 (297276_1_supp_6694955_rrs73y.pdf).** Screenshots of the ACME tool. (A) is the image in the DAPI channel, (B) FISH channel, and (C) the autofluorescence channel. DAPI positive objects have a light-blue, FISH positive a green, and auto-fluorescent particles

a red outline. Red box is a zoom-in on an example of a FISH positive cell with two local DAPI maxima (i.e., a dividing cell). Yellow circle is around an algae cell, yellow arrow points towards debris. Images are an example of 20th April and the samples were hybridized with the SAR11 mix. Each field of view is 1388x1040 pixel and each pixel has a height and width of 0.106 µm.

SUPPLEMENTAL FILE 1 (297276_1_supp_6684536_rrs7qx.docx). Supplementary Material and Methods and Results.

## Open Peer Review

PEER REVIEW HISTORY (review-history.pdf). An accounting of the reviewer comments and feedback.

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
