## [Reviewer comments · mSystems]

***In situ* cell division and mortality rates of SAR11, SAR86, *Bacteroidetes*, and *Aurantivirga* during phytoplankton blooms reveal differences in population controls**

Jan Brüwer, Luis Orellana, Chandni Sidhu, Helena Klip, Cédric Meunier, Maarten Boersma, Karen Wiltshire, Rudolf Amann, and Bernhard Fuchs

Corresponding Author(s): Bernhard Fuchs, Max Planck Institute for Marine Microbiology

Review Timeline:

Submission Date:	December 21, 2022
Editorial Decision:	February 5, 2023
Revision Received:	March 20, 2023
Accepted:	March 21, 2023

Editor: Jean-Baptiste Raina

Reviewer(s): The reviewers have opted to remain anonymous.

Transaction Report:

DOI: <https://doi.org/10.1128/msystems.01287-22>

February 5, 2023

Dr. Bernhard M Fuchs
Max Planck Institute for Marine Microbiology
Molecular Ecology
Celsiusstr. 1
Bremen 28359
Germany

Re: mSystems01287-22 (*In situ* cell division and mortality rates of SAR11, SAR86, *Bacteroidetes*, and *Aurantivirga* during phytoplankton blooms reveal differences in population controls)

Dear Dr. Bernhard M Fuchs:

Thank you for submitting your manuscript to mSystems. We have completed our review and I am pleased to inform you that, in principle, we expect to accept it for publication in mSystems. However, acceptance will not be final until you have adequately addressed the reviewer comments.

Both reviewers find your work to be of high quality, but feel that the Discussion needs to be extended. In addition, the maximum number of supplementary figure allowed for publication is 10. Please combine some of your supplementary figures in your resubmitted version.

Below you will find instructions from the mSystems editorial office and comments generated during the review.

Preparing Revision Guidelines

Sincerely,

Jean-Baptiste Raina

Editor, mSystems

Reviewer comments:

Reviewer #1 (Comments for the Author):

This paper reports on the net and gross growth rates of some relevant bacterioplankton groups in the North Sea. The authors use image analyses of FISH-labelled cells in combination with in situ sampling and dilution cultures, as well as metagenomic analyses of MAGs to describe the seasonality of the growth of these groups. It is a very nice study that shows the correspondence between cell volume, ribosome relative content and frequency of dividing cells. I'm very sympathetic with the approach chosen, and the use of microscopy, something that has almost been abandoned by the deluge of molecular data, and the higher fascination that computer work has in our students in front of microscopic work. My congrats to the authors. Please, get it published soon, as I want my students to read it! (and learn that methods of the 70s can still be useful).

In spite of my general great consideration of this work, there are a few things that should somehow be discussed:

- A lot of the conclusions are based on the relationship in Fig S5 (which I believe is relevant enough to have it as main figure). Yet the SAR86 relationship is pretty weak... and in line 158 you make inferences about the growth of SAR86 that would be completely different if all the data in this figure (except the kinda outlier of SAR86) were fitted the same line. The max growth of SAR86 (0.5) would have been 1.5 and your conclusion completely different.
- I feel uneasy with the Net vs Gross growth rate comparisons. The reasons are that your GGR are kinda "instantaneous", while your net growth rates are based on a time-span of ca 5 days (l. 177). This implicates that the net values of growth are low and consequently that the high growth rates are compensated by high loss rates. While I have no doubts on the loss rates being important... I would be more confident had you looked at daily data (if available). Needham & Fuhrman showed data suggesting that some bacteria change a lot in abundance in consecutive days... indicating that true in situ (net) rates of growth are also high. In brief: there is a temporal component when you compute rates of growth: If you use the changes from year to year, net growth is minimum. Which is the right temporal scale to measure net growth?
- I really didn't get what the GRiD or gRodon data add to your work... Your paragraph in lines >248 is basically saying "don't loose time doing this". You also add in the abstract a line (l. 39-40) which in fact isn't supported by the data. You might have added this in response to some bioinformatics freak reviewer, but your paper DOES NOT support this point (at least to the point of adding it to the abstract).
- The cell sizes of SAR11 (0.1-0.18 μm^3 , l. 131) are extremely high (compare to Zhao et al AEM 2016). I understand you have a different species, but this should be discussed: is it because of the IA system you used? In any case, whether the absolute cell sizes are real or not doesn't really matter for this ms. but the discussion is mandatory. Similarly, those of SAR86 and Bacteroidetes (lines >286) are much larger than previously measured. A discussion beyond what you say in lines 294-300 in which you simply say "the volumes were much larger" would be nice..

Smaller comments:

- line 146. Unsure I understand this comment here. In principle, FDC should be a much better estimate of growth rate than volume or ribosome content.
- line 154. Not sure I buy the multiple regression argument. You don't need it: use the data for each bacterial group to compute group-specific relationships. You don't need the multiple regression... unless your stats for SAR86 didn't hold. In fact, I doubt about the high point - GR 0.5 with FDC 9%??-). If you remove it you can compute a single line that explains all the data (yet the slope is lower then for all groups).
- Fig 1. I'm not very fond of these combined figures in which half of the panels share the same axis (time in this case), and the other half are X-Y plots of two variables. I find this very confusing and I believe deters understanding. I would strongly suggest to divide this figure into 2, panels A-D, vs panels E-H. And if the journal doesn't allow more figures (why this limit in times of ejournals?), then combine the panels in two groups with a heading.
- Figs 2 and 3, Figs S2, S3, S4. Please place them in the right orientation so that the reviewer (and reader) doesn't have to twist the head to read the text!
- Fig S5. Why FDC on the Y axis? You will be calculating GR from FDC, so, FDC makes more sense in the X, no?
- l. 155. I hate reporting P values like this ($4 \cdot 10^{-5}$). This is not informative. Much better to report them as $P < 0.01$ or $P < 0.0001$ if you wish.
- l. 183 and Fig S6. I see here relatively poor group specific relationships... and very variable between them (some negative). Shouldn't this be discussed?
- Fig S7 is unreadable. Being A supplementary one, I would split it into more readable different figures.
- Fig S8. Maybe the abundances can be stacked, but the GRiD values????? Not much sense to me... (stacked bar graphs are unreadable).
- l. 225. I do not know what an "interactive model" is.
- l. 225. Do you really see a positive correlation of GRiD vs FDC in this figure? Maybe in Fig S9, but unclear (if you don't show an

X-Y graph).

- l. 228. I am missing an integration of these data with the previous results about the MAGs. I can't know which one is which based on this information. Also, I believe you should give more background info about the abundance of the identified MAGs (Fig S7A-D). Is r116 the one dominating? (it looks like is r27). This is relevant as your SAR were large and grew fast.... But if the reader can't tell which one is which...
- l. 228/Fig 4. I'm unsure this helps the ms. The values identified are way off what you measure as in situ growth rates (look similar just because of h-1 or d-1), and basically only indicate that SAR11/SAR86 (in bulk) are more likely to grow slower than the other groups... You don't even discuss it in l. 258.
- l. 239. That comparison, not explained (reference to the SI doesn't help) confuses more than anything else. I would say, in the M&M section that your CARDFish data was good, and avoid any further discussion.
- l. 242. What is more robust? That explains better what? That has a higher correlation? Be specific.
- l. 246. See my comment above.
- l. >260. In this paragraph you make the point that resources play a larger role than temp in determining growth rate. However, the discussion doesn't go further, when it could be possible to play with the measured data, chl data and temperature. An old paper (White et al. 1991, *Mic Ecol*) provides models of these three variables. Maybe you should discuss also in light of Lopez-Urrutia et al. 2007 *Ecology*. There might be more newer references...
- l. 306. This discussion could be better by providing some of the light data. I bet your North Sea site is cloud-covered most of the time. What's the light regime? I'm sure it is measured in the Helgoland LTER!
- l. 314. As you say, SAR11 specific grazing hasn't been shown, and all indications concur in that SAR11 are under low grazing pressure. But what about viruses? The Ferrera and Sanchez papers (see the one in *Sci Rep* 2020) specifically tested this. What were their conclusions? Since you have metagenomes... are there signals of SAR11 viruses, such as those studied by the Martinez-Hernandez lab?
- l. 316. I agree that some SAR11 species are likely not typical oligotrophs. How similar are your MAGs to other SAR11? Why Fig S12 is not even cited and discussed here?
- The discussion on SAR11 could benefit of citing Sanchez et al 2020 *Sci Rep*, where they show the seasonally changing effect of viruses and predators on SAR growth rates. These authors show SAR11 to also grow well above 1 d-1 when predators are removed in the NW Med sea.
- Suppl Info. I'm indeed surprised with the material here included, particularly with the non-methods results here included. First the comparison between CARDFISH and the tetra labelled probes... Weren't you guys, the inventors of the method dubious of CardFish????? Them, why there is no reference to the taxonomy of SAR11 in the main text? Isn't it relevant? I'm saying this because if you explain the taxonomy it is because you think it might be relevant, right? Or it is not? If it is not, why to include it?

Reviewer #2 (Comments for the Author):

The authors present a relevant and consistent study on an essential process in population ecology. They rigorous work nicely demonstrate the importance of in situ cytometry studies to measure cell division and highlight the importance experimental work to determine net growth rates and estimate loss terms (top-down factors). The results are sound and the manuscript is well written. Nevertheless, I have some comments and concerns that might help to improve the manuscript. Overall the discussion does not properly discuss important ecological aspects. The author should take advantage of their dataset to discuss more deeply the observed differences in FDC, cell division and growth among taxa, and how mortality and division rates relate with environmental variables. I wonder if the authors have data on inorganic and organic nutrients. In lines 259-265, the authors mention the limited role that temperature seems to play in population control in the study area. However, it seems that the temperate correlate very well with bacterial abundance in 2018 (Fig. S1). They mention some other factors that may be important but they do not provide additional biotic or abiotic data (only temperature and chlorophyll) data, and therefore this aspect remains poorly resolved. On the other hand, the author could enrich their discussion about the tight coupling they found between mortality and cell division rates. This is a very interesting finding which definitively deserves more discussion. Another result that deserves more discussion is the fact that the correlation between mortality and the estimated grazing seems to apply only for SAR11. An important limitation is that the experimental design does not allow to estimate growth (cell division) rates, as it only allows to estimate the protist-free growth rate (but not the protist+viruses free growth rate). This methodological limitation of the dilution technique should be properly addressed. On the other hand, as the authors have estimations of FDC also for 2020, I suggest showing these data in figure 2, and compare both sampling periods. They can remove cell division rates from figure 2 as they are already represented in figure 3. In the same way, the authors can calculate net growth rates also for 2018 and compare both sampling periods.

Specific comments

- Line 155. It is quite confusing how the equation is expressed. It is not clear what variable is "FISH-probe", I can guess what the authors mean, but I suggest to clarify this calculation.
- Line 184. Please clarify which data were used for such correlation. In fire S6 it is clear that this correlation is positive only using SAR11 data. I assume that the R2 and the p value correspond to the correlation including all data. Please clarify.
- Lines 212-216. Please revise figure S7 as the data for Bacteroidetes are not visible in the plots, and thus, this text could not be followed.
- Line 209 and 217. Please spell out the acronyms MAG and GRID the first time they appear in the text.
- Line 226. Please indicate the p-value.
- Lines 298-299. It is not clear what the authors mean here by lower plasticity. Compared with what?

Lines 311. It is not true that the estimated SAR11 net growth rates are one order magnitude higher than the other three taxa. They are in the same range.

Line 426. This amount of ADN seems too high for only 1 l of filtered sample.

Review of manuscript entitled “In situ cell division and mortality rates of SAR11, 1 SAR86, Bacteroidetes, and Aurantivirga during phytoplankton blooms reveal differences in population controls”

The authors present a relevant and consistent study on an essential process in population ecology. They rigorous work nicely demonstrate the importance of in situ cytometry studies to measure cell division and highlight the importance experimental work to determine net growth rates and estimate loss terms (top-down factors). The results are sound and the manuscript is well written. Nevertheless, I have some comments and concerns that might help to improve the manuscript. Overall the discussion does not properly discuss important ecological aspects. He author should take advantage of their dataset to discuss more deeply the observed differences in FDC, cell division and growth among taxa, and how mortality and division rates relate with environmental variables. I wonder if the authors have data on inorganic and organic nutrients. In lines 259-265, the authors mention the limited role that temperature seems to play in population control in the study area. However, it seems that the temperate correlate very well with bacterial abundance in 2018 (Fig. S1). They mention some other factors that may be important but they do not provide additional biotic or abiotic data (only temperature and chlorophyll) data, and therefore this aspect remains poorly resolved. On the other hand, the author could enrich their discussion about the tight coupling they found between mortality and cell division rates. This is a very interesting finding which definitively deserves more discussion. Another result that deserves more discussion is the fact that the correlation between mortality and the estimated grazing seems to apply only for SAR11. An important limitation is that the experimental design does not allow to estimate growth (cell division) rates, as it only allows to estimate the protist-free growth rate (but not the protist+viruses free growth rate). This methodological limitation of the dilution technique should be properly addressed. On the other hand, as the authors have estimations of FDC also for 2020, I suggest showing these data in figure 2, and compare both sampling periods. They can remove cell division rates from figure 2 as they are already represented in figure 3. In the same way, the authors can calculate net growth rates also for 2018 and compare both sampling periods.

Specific comments

Line 155. It is quite confusing how the equation is expressed. It is not clear what variable is “FISH-probe”, I can guess what the authors mean, but I suggest to clarify this calculation.

Line 184. Please clarify which data were used for such correlation. In fire S6 it is clear that this correlation is positive only using SAR11 data. I assume that the R2 and the p value correspond to the correlation including all data. Please clarify.

Lines 212-216. Please revise figure S7 as the data for Bacteroidetes are not visible in the plots, and thus, this text could not be followed.

Line 209 and 217. Please spell out the acronyms MAG and GRID the first time they appear in the text.

Line 226. Please indicate the p-value.

Lines 298-299. It is not clear what the authors mean here by lower plasticity. Compared with what?

Lines 311. It is not true that the estimated SAR11 net growth rates are one order magnitude higher than the other three taxa. They are in the same range.

Line 426. This amount of ADN seems too high for only 1 l of filtered sample.

Our comments are in green.

Paragraphs of the revised manuscript are in blue and italics.

Please note that supplementary tables are under embargo until publication. They can be accessed via: <https://figshare.com/s/4b796886600f5c2636d7>

Dear Dr. Bernhard M Fuchs:

Thank you for submitting your manuscript to mSystems. We have completed our review and I am pleased to inform you that, in principle, we expect to accept it for publication in mSystems. However, acceptance will not be final until you have adequately addressed the reviewer comments.

Both reviewers find your work to be of high quality, but feel that the Discussion needs to be extended. In addition, the maximum number of supplementary figure allowed for publication is 10. Please combine some of your supplementary figures in your resubmitted version.

Below you will find instructions from the mSystems editorial office and comments generated during the review.

Preparing Revision Guidelines

ASM policy requires that data be available to the public upon online posting of the article, so please verify all links to sequence records, if present, and make sure that each number retrieves the full record of the data. If a new accession number is not

linked or a link is broken, provide production staff with the correct URL for the record. If the accession numbers for new data are not publicly accessible before the expected online posting of the article, publication of your article may be delayed; please contact the ASM production staff immediately with the expected release date.

Sincerely,

Jean-Baptiste Raina

Editor, mSystems

Journals Department
Reviewer comments:

Reviewer #1 (Comments for the Author):

This paper reports on the net and gross growth rates of some relevant bacterioplankton groups in the North Sea. The authors use image analyses of FISH-labelled cells in combination with in situ sampling and dilution cultures, as well as metagenomic analyses of MAGs to describe the seasonality of the growth of these groups. It is a very nice study that shows the correspondence between cell volume, ribosome relative content and frequency of dividing cells. I'm very sympathetic with the approach chosen, and the use of microscopy, something that has almost been abandoned by the deluge of molecular data, and the higher fascination that computer work has in our students in front of microscopic work. My congrats to the authors. Please, get it published soon, as I want my students to read it! (and learn that methods of the 70s can still be useful).

Thank you for your kind comment! This is very encouraging!

In spite of my general great consideration of this work, there are a few things that should somehow be discussed:

- A lot of the conclusions are based on the relationship in Fig S5 (which I believe is relevant enough to have it as main figure). Yet the SAR86 relationship is pretty weak... and in line 158 you make inferences about the growth of SAR86 that would be completely different if all the data in this figure (except the kindda outlier of SAR86) were fitted the same line. The max growth of SAR86 (0.5) would have been 1.5 and your conclusion completely different.

We agree! This is a very important point. We have updated the results accordingly and discuss these results with a grain of salt. We answered this question in detail below.

- I feel uneasy with the Net vs Gross growth rate comparisons. The reasons are that your GGR are kindda "instantaneous", while your net growth rates are based on a time-span of ca 5 days (l. 177). This implicates that the net values of growth are low and consequently that the high growth rates are compensated by high loss rates. While I have no doubts on the loss rates being important... I would be more confident had you looked at daily data (if available). Needham & Fuhrman showed data suggesting that some bacteria change a lot in abundance in consecutive days... indicating that true in situ (net) rates of growth are also high. In brief: there is a temporal component when you compute rates of growth: If you use the changes from year to year, net growth is minimum. Which is the right temporal scale to measure net growth?

Population abundances can change within short time periods as short as days (e.g., Needham & Fuhrman) or even within 24 hours. In our manuscript, we have a rather broad view on the cell abundances and assess the changes in cell abundances over multiple days to weeks. Therefore, we think it is acceptable to remove some noise from the data by averaging over this time-span of 5 days.

We have reduced the time-span to three consecutive sampling points and are happy to provide the same figure with net growth rates - and mortality rates - between individual data points for your reference below.

We hope you agree that by comparing this figure to the original figure you will find the differences introduced with the different time-spans are negligible.

- I really didn't get what the GRiD or gRodon data add to your work... Your paragraph in lines >248 is basically saying "don't loose time doing this". You also add in the abstract a line (l. 39-40) which in fact isn't supported by the data. You might have added this in response to some bioinformatics freak reviewer, but your paper DOES NOT support this point (at least to the point of adding it to the abstract).

Thank you very much. We very much agree with the reviewer and have removed any mentioning of metagenome-derived estimates from our abstract. We have also re-worked the discussion about these bioinformatic methods:

We computed GRiD (31), which were highly susceptible to the mapping tools that were used, not yielding any reproducible results. Therefore, we cannot support using the GRiD algorithm at this developmental stage.

We are convinced, however, that gRodon could still be useful to identify oligo- or copiotrophic lifestyles for individual taxa. Whether this has any (ecologic) implications remains to be explored in future studies, though.

- The cell sizes of SAR11 (0.1-0.18 μm^3 , l. 131) are extremely high (compare to Zhao et al AEM 2016). I understand you have a different species, but this should

be discussed: is it because of the IA system you used? In any case, whether the absolute cell sizes are real or not doesn't really matter for this ms. but the discussion is mandatory. Similarly, those of SAR86 and Bacteroidetes (lines >286) are much larger than previously measured. A discussion beyond what you say in lines 294-300 in which you simply say "the volumes were much larger" would be nice..

We agree. We have now included a disclaimer with methodological limitations in the discussion, which give possible reasons why our cell sizes are most likely overestimations. Most likely our cells are spread flat on the filter surface, losing some of its height. Therefore, our 3D models of cells volumes might overestimate in the third dimension.

We added this paragraph to the discussion:

Our image cytometry approach had some limitations regarding cell volume measurements and dilution experiments. First, cells are filtered onto polycarbonate filter and might lose some of their height due to fixation. Therefore, our 3D models of cells volumes most likely somewhat overestimate in the third dimension. Second, cell volume measurements are derived from a CARD amplification signal, which often seem to overshadow the cell boundaries and hence overestimate the cell dimensions. Additionally, object identification and volume measurement were both done on the same signal. The thresholds to identify a cell, immediately influence the cell size and volume estimates (36). Taken together, this could contribute to an overestimation of cell volume measurements. Nevertheless, this should not affect comparisons of cell volumes within this study. Finally, our dilution experiments did not exclude phage-free cell division rates, as other studies have done (33). The dilutions were prepared with 0.2 μm filtered water, which is larger than most phages.

Smaller comments:

- line 146. Unsure I understand this comment here. In principle, FDC should be a much better estimate of growth rate than volume or ribosome content.

We have rephrased and split the sentence into two for clarification.

- line 154. Not sure I buy the multiple regression argument. You don't need it: use the data for each bacterial group to compute group-specific relationships. You don't need the multiple regression... unless your stats for SAR86 didn't hold. In fact, I doubt about the high point - GR 0.5 with FDC 9%??-). If you remove it you can compute a single line that explains all the data (yet the slope is lower than for all groups).

Thank you very much for these thoughts!

First, although the mentioned data point is statistically an outlier (Grubb's Test. $P < 0.002$), we will not remove it because it would be tantamount to "cherry

picking” to leave this data point out. We do agree with you and reviewer 2, that the interpretation of this needs to be addressed in the discussion, which has been done.

Second, statistically the null hypothesis is (a) that FDC and cell division rate are not correlated and (b) that the relationship between the two is not taxon-specific. We reject both with a multiple regression. We have included this argument in the manuscript:

*We used multiple linear regressions with the null hypothesis that a) FDC (“FDC”) is independent of experimentally-derived cell division rates (“ μ ”) and b) this relationship is independent of the assessed taxon (“taxon”); formula: $FDC \sim \mu * taxon$). We rejected both null hypotheses ($R^2 = 0.86$; $p < 0.0001$) and could calculate taxon-specific cell division rates from the FDC across the 2020 spring bloom (Fig. S3A, Table S2).*

Third, the data for FDC is normally distributed (Shapiro-Wilk normality Test for FDC: $p=0.18$, for cell division rate: $p=0.077$), if we use the multiple regression model. This is not the case (e.g., for SAR86), when done separately.

Last, we initially did test all four taxa individually. The SAR86 FDC to cell division rate is statistically significant ($p=0.011$, $R^2=0.89$). Additionally, the derived linear models have the same formula for all taxa, irrespective, whether individual models or the multiple regression models are chosen.

- Fig 1. I'm not very fond of these combined figures in which half of the panels share the same axis (time in this case), and the other half are X-Y plots of two variables. I find this very confusing and I believe deters understanding. I would strongly suggest to divide this figure into 2, panels A-D, vs panels E-H. And if the journal doesn't allow more figures (why this limit in times of ejournals?), then combine the panels in two groups with a heading.

We have adjusted our figures to your suggestions.

- Figs 2 and 3, Figs S2, S3, S4. Please place them in the right orientation so that the reviewer (and reader) doesn't have to twist the head to read the text!

We have adjusted our figures to your suggestions and combined S2, S3, and S4 into one figure.

- Fig S5. Why FDC on the Y axis? You will be calculating GR from FDC, so, FDC makes more sense in the X, no?

Thank you for this thought and it feels like the chicken and egg question. We consider the independent variable to be the cell division rate and the FDC is the depending on the cell division.

- l. 155. I hate reporting P values like this ($4 \cdot 10^{-5}$). This is not informative. Much better to report them as $P < 0.01$ or $P < 0.0001$ if you wish.

Done. We have changed this throughout the manuscript.

- l. 183 and Fig S6. I see here relatively poor group specific relationships... and very variable between them (some negative). Shouldn't this be discussed?

First of all, as we have reduced the sliding window to calculate net growth and consequently mortality, we have also updated this figure.

We agree that this needed more attention. In our opinion, the most important point is that most data are around the 1:1 ratio, which indicates that our calculated mortality rates are in-line with measured grazing rates and that virus lysis seem to be less important.

- Fig S7 is unreadable. Being a supplementary one, I would split it into more readable different figures.

We understand it is barely readable. As you have pointed out in a previous comment, the bioinformatic insights are vague and not informative at all. We have decided to keep this figure and add another supplementary table with all raw values for the interested reader.

- Fig S8. Maybe the abundances can be stacked, but the GRiD values????? Not much sense to me... (stacked bar graphs are unreadable).

We agree! As indicated above, we now supply a table with more information for the interested reader.

- l. 225. I do not know what an "interactive model" is.

We used multiple linear regression models, similar to the FDC and cell division rate ratio. We have updated the section to make this clear.

- l. 225. Do you really see a positive correlation of GRiD vs FDC in this figure? Maybe in Fig S9, but unclear (if you don't show an X-Y graph).

Fig. S5 (Fig. S7 in the original publication), panels I-L include the suggested X-Y plots. We reported the corresponding statistics of the positive correlation. Nevertheless, we have discussed the limitations of GRiD values and do not advocate using this software for our purposes.

- l. 228. I am missing an integration of these data with the previous results about

the MAGs. I can't know which one is which based on this information. Also, I believe you should give more background info about the abundance of the identified MAGs (Fig S7A-D). Is r116 the one dominating? (it looks like is r27). This is relevant as your SAR were large and grew fast.... But if the reader can't tell which one is which...

We have added further details. For detailed list of, e.g., Bacteroidetes MAGs, we refer to the supplementary material.

- l. 228/Fig 4. I'm unsure this helps the ms. The values identified are way off what you measure as in situ growth rates (look similar just because of h-1 or d-1), and basically only indicate that SAR11/SAR86 (in bulk) are more likely to grow slower than the other groups... You don't even discuss it in l. 258.

We have updated Fig. 4, which now displays maximum cell division rates, rather than minimal doubling time. The figure shows the genomic potential and a predicted value. The predicted values are all much faster than our measured values. Hence, we cannot disprove this bioinformatic pipeline (gRodon). It may or may not be useful for other users.

We included a paragraph in the discussion:

The gRodon results were in line with our assumptions that SAR11 and SAR86 can be considered oligotrophs and Aurantivirga a copiotroph. Bacteroidetes being heterogeneous, with the majority of clades putatively slow growing, confirms previous findings of few actively growing Bacteroidetes clades during phytoplankton blooms (5). All experimental cell division rates were slower than gRodon-predicted genomic potentials for maximum cell division rates, which indicates that – on a community level – none of the assessed groups divides to their full capacity.

- l. 239. That comparison, not explained (reference to the SI doesn't help) confuses more than anything else. I would say, in the M&M section that your CARDFish data was good, and avoid any further discussion.

We have amended the corresponding section accordingly.

- l. 242. What is more robust? That explains better what? That has a higher correlation? Be specific.

Thanks. We have rephrased the entire paragraph to be more precise and less ambiguous.

- l. 246. See my comment above.

We have amended the section to be more precise.

- l. >260. In this paragraph you make the point that resources play a larger role than temp in determining growth rate. However, the discussion doesn't go further, when it could be possible to play with the measured data, chl data and temperature. An old paper (White et al. 1991, *Mic Ecol*) provides models of these three variables. Maybe you should discuss also in light of Lopez-Urrutia et al. 2007 *Ecology*. There might be more newer references...

Thank you very much for this comment! We realized that the entire paragraph was ambiguous. We have re-worked and updated it carefully.

Under constant substrate and nutrient conditions, cell division and mortality rates are both temperature dependent (40, 41). This is also known for bacteria in environmental samples (e.g., 42, 43) but only partly visible in our case (Fig. S1, Table S5). For example, though the temperature increased between April and May 2020 from 9.9°C to 11.4° (Fig. S1B), the cell division rates of Bacteroidetes and Aurantivirga decreased in May and SAR11 cell division rates fell to pre-bloom levels in mid-April and end of May. Other than temperature, the bacterial communities are shaped by phytoplankton-derived organic matter (1, 2). Inorganic nutrients such as nitrate, ammonium, phosphate, and silicate, which are tightly monitored at the LTER Helgoland, are negatively correlated with FDC (Table S5) (2020 data: 9, 2018 data: 32). However, these nutrients are directly taken up and depleted by phytoplankton and are, thus, only indirectly correlated with FDC without causation (2, 32).

- l. 306. This discussion could be better by providing some of the light data. I bet your North Sea site is cloud-covered most of the time. What's the light regime? I'm sure it is measured in the Helgoland LTER!

Unfortunately, this is not part of the Helgoland LTER. However, we have taken a look at remote sensing data (Aqua Modis) for the German Bight. It is now included in the manuscript (Fig. S1, S9, see below).

The light intensity (PAR) increases over the spring bloom. There could be a threshold of $\sim 25 \text{ Einstein m}^{-2} \text{ d}^{-1}$ for SAR11 to increase their activity.

Hence, increasing light intensities during the spring blooms could support growth by fuelling energy-dependent transport albeit only SAR11 cells seemed to have benefitted from this. Above $\sim 25 \text{ Einstein m}^{-2} \text{ d}^{-1}$, SAR11 cell division was increased gradually (Fig. S1, S9), which could potentially be considered as a threshold in our case to obtain enough energy for increased activity. Previous incubations detected increased proteorhodopsin-derived activity in SAR11 after incubations with $36 \text{ Einstein m}^{-2} \text{ d}^{-1}$ (54).

- l. 314. As you say, SAR11 specific grazing hasn't been shown, and all indications concur in that SAR11 are under low grazing pressure. But what about viruses? The

Ferrera and Sanchez papers (see the one in Sci Rep 2020) specifically tested this. What were their conclusions? Since you have metagenomes... are there signals of SAR11 viruses, such as those studied by the Martinez-Hernandez lab?

Thank you. We have expanded our discussion on the influence of viruses on the SAR11 community. The query of the metagenomes for SAR11 viruses is not trivial and could be done in the future. However, it is not in the scope of this study.

- l. 316. I agree that some SAR11 species are likely not typical oligotrophs. How similar are your MAGs to other SAR11? Why Fig S12 is not even cited and discussed here?

We agree! We have updated the corresponding sections in the manuscript and now mention Fig. S4 (Fig. S12 in the original submission) in the main text:

SAR11 was represented by 5 MAGs, of which 4 belong to the open ocean clade 1a.1 and MAG r31 to clade 3 (Fig. S4) (34, 35).

- The discussion on SAR11 could benefit of citing Sanchez et al 2020 Sci Rep, where they show the seasonally changing effect of viruses and predators on SAR growth rates. These authors show SAR11 to also grow well above 1 d⁻¹ when predators are removed in the NW Med sea.

Thank you very much for this valuable input. We have included it in our discussion for the seasonal effect of viruses and showing fast SAR11 growth rates.

- Suppl Info. I'm indeed surprised with the material here included, particularly with the non-methods results here included. First the comparison between CARDFISH and the tetra labelled probes... Weren't you guys, the inventors of the method dubious of CardFish?????

As you suggested in an earlier comment, we have shortened this section and now mention our quality control step in the material and methods.

Them, why there is no reference to the taxonomy of SAR11 in the main text? Isn't it relevant? I'm saying this because if you explain the taxonomy it is because you think it might be relevant, right? Or it is not? If it is not, why to include it?

Yes, it is important! We have updated this accordingly (see comments above).

Reviewer #2 (Comments for the Author):

The authors present a relevant and consistent study on an essential process in population ecology. Their rigorous work nicely demonstrates the importance of in situ cytometry studies to measure cell division and highlights the importance of experimental work to determine net growth rates and estimate loss terms (top-down factors). The results are sound and the manuscript is well written. Nevertheless, I have some comments and concerns that might help to improve the manuscript. Overall, the discussion does not properly discuss important ecological aspects. The author should take advantage of their dataset to discuss more deeply the observed differences in FDC, cell division and growth among taxa, and how mortality and division rates relate with environmental variables. I wonder if the authors have data on inorganic and organic nutrients.

Thank you for your concerns. We have re-worked the discussion to include further ecological aspects. We now take a closer look at photosynthetically active radiation (PAR; Fig. S1), temperature (Fig. S1, Table S5), and some inorganic nutrients. However, we know that nutrients (nitrate, silica, phosphate; Table S5) become limited during spring blooms, as they are being used by the phytoplankton. Hence, correlation would not indicate causation.

In lines 259-265, the authors mention the limited role that temperature seems to play in population control in the study area. However, it seems that the temperature correlates very well with bacterial abundance in 2018 (Fig. S1).

We agree. This paragraph was ambiguous and has been reworked entirely.

Our statement was mis-leading. Instead of 'population control', we explored the potential effect of temperature on the cell division activity, assessed by FDC. The manuscript now reads:

Under constant substrate and nutrient conditions, cell division and mortality rates are both temperature dependent (40, 41). This is also known for bacteria in environmental samples (e.g., 42, 43) but only partly visible in our case (Fig. S1, Table S5). For example, though the temperature increased between April and May 2020 from 9.9°C to 11.4°C (Fig. S1B), the cell division rates of Bacteroidetes and Aurantivirga decreased in May and SAR11 cell division rates fell to pre-bloom levels in mid-April and end of May. Other than temperature, the bacterial communities are shaped by phytoplankton-derived organic matter (1, 2). Inorganic nutrients such as nitrate, ammonium, phosphate, and silicate, which are tightly monitored at the LTER Helgoland, are negatively correlated with FDC (Table S5) (2020 data: 9, 2018 data: 32). However, these nutrients are directly taken up and depleted by phytoplankton and are, thus, only indirectly correlated with FDC without causation (2, 32).

They mention some other factors that may be important but they do not provide additional biotic or abiotic data (only temperature and chlorophyll) data, and therefore this aspect remains poorly resolved.

Please see our comments above. Additionally, we have looked at PAR (Fig. S1, S9), which might play a role in the high SAR11 activity before the phytoplankton bloom.

Hence, increasing light intensities during the spring blooms could support growth by fuelling energy-dependent transport albeit only SAR11 cells seemed to have benefitted from this. Above ~ 25 Einstein $m^{-2} d^{-1}$, SAR11 cell division was increased gradually (Fig. S1, S9), which could potentially be considered as a threshold in our case to obtain enough energy for increased activity. Previous incubations detected increased proteorhodopsin-derived activity in SAR11 after incubations with 36 Einstein $m^{-2} d^{-1}$ (54).

On the other hand, the author could enrich their discussion about the tight coupling they found between mortality and cell division rates. This is a very interesting finding which definitively deserves more discussion.

Done.

Another result that deserves more discussion is the fact that the correlation between mortality and the estimated grazing seems to apply only for SAR11.

We agree! We have amended the corresponding section and provide additional information. While the individual correlations seem to be scattered, it is noteworthy that the data generally follows the 1:1 ratio.

Nevertheless, data from all taxa combined followed the 1:1 ratio or calculated mortality was larger than grazing. This indicates that our calculated mortality rates can to a large extent be explained by grazing, with few cases where, for example, viral lysis might play an important role. Similarly, Sanchez et al. (33) found in a recent study that mortality due to grazers was larger than viral lysis, across multiple seasons and bacterial taxa.

...

We validated our derived mortality rates against the grazing rates obtained from the dilution experiments, which largely followed a 1:1 ratio. A 1:1 ratio would mean that all mortality is due to grazing. Our results reveal that despite several cell divisions per day, mortality diminishes the increase in cell abundance. In other words, cell division rates in the environment are higher than anticipated, however mortality removes >90% of newly produced bacterial biomass each day.

An important limitation is that the experimental design does not allow to estimate growth (cell division) rates, as it only allows to estimate the protist-free growth rate (but not the protist+viruses free growth rate). This methodological limitation of the dilution technique should be properly addressed.

We agree. We have added a paragraph to acknowledge the major limitations to this study:

Our image cytometry approach had some limitations regarding cell volume measurements and dilution experiments. (...) Finally, our dilution experiments did not exclude phage-free cell division rates, as other studies have done (33). The dilutions were prepared with 0.2 μm filtered water, which is larger than most phages.

On the other hand, as the authors have estimations of FDC also for 2020, I suggest showing these data in figure 2, and compare both sampling periods. They can remove cell division rates from figure 2 as they are already represented in figure 3.

Done. We agree it was redundant.

In the same way, the authors can calculate net growth rates also for 2018 and compare both sampling periods.

Thank you for this remark! At this stage, we refrain from calculating cell division rates for 2018, based on the 2020 data. We do not know, whether there is a 'general' translation for growth rates, yet, or if calibrations with dilution experiments are always needed. This needs to be explored in future studies.

We already stated in the original submission: "*We could only assess relative growth activity changes by studying FDC in 2018, as calibrations of FDC with dilution experiments were only done in 2020.*"

Specific comments

Line 155. It is quite confusing how the equation is expressed. It is not clear what variable is "FISH-probe", I can guess what the authors mean, but I suggest to clarify this calculation.

The term was changed to "taxon" instead of FISH-probe. We re-phrased the text to make it less ambiguous:

*We used multiple linear regressions with the null hypothesis that a) FDC ("FDC") is independent of experimentally-derived cell division rates (" μ ") and b) this relationship is independent of the assessed taxon ("taxon"; formula: $FDC \sim \mu * \text{taxon}$). We rejected both null hypotheses ($R^2 = 0.86$; $p < 0.0001$) and could calculate taxon-specific cell division rates from the FDC across the 2020 spring bloom (Fig. S3A, Table S2)*

Line 184. Please clarify which data were used for such correlation. In figure S6 it is clear that this correlation is positive only using SAR11 data. I assume that the R2 and the p value correspond to the correlation including all data. Please clarify.

Done. We have updated the section:

*We compared these calculated mortality rates to grazing rates, which were determined in the dilution experiments. Both were significantly correlated in a multiple regression model of grazing ~ mortality*taxon (R2 = 0.86; p = 0.002; Fig. S3B).*

Lines 212-216. Please revise figure S7 as the data for Bacteroidetes are not visible in the plots, and thus, this text could not be followed.

As mentioned above for reviewer #1, we updated our discussion about the usefulness of GRiD values for our purpose. We conclude they are not. Nevertheless, we now include a supplementary table for the interested reader (Table S3), and kept the figure to provide a broad overview.

Line 209 and 217. Please spell out the acronyms MAG and GRID the first time they appear in the text.

Done.

Line 226. Please indicate the p-value.

Done. It now reads:

The determined GRiD values correlated positively to FDC, with a taxon-specific interaction term, though with high variance ($p < 0.0001$, $R2 = 0.12$; Fig. S7).

Lines 298-299. It is not clear what the authors mean here by lower plasticity. Compared with what?

We have updated this paragraph which now reads:

Due to the assumed slower cell division rates of SAR11, we hypothesized less variation in cell volumes and ribosomal content compared to putatively faster growing Bacteroidetes and Aurantivirga (52). While generally lower, the ribosomal content of SAR11 cells fluctuated comparable to the three other taxa.

Lines 311. It is not true that the estimated SAR11 net growth rates are one order magnitude higher than the other three taxa. They are in the same range.

Corrected, it now reads:

Net growth was almost an order of magnitude lower than the cell division rate (...).

Line 426. This amount of DNA seems too high for only 1 l of filtered sample.

Correct! These are ng/ μ l of DNA per L - not μ g/ μ l. Changed accordingly.

March 21, 2023

Dr. Bernhard M Fuchs
Max Planck Institute for Marine Microbiology
Molecular Ecology
Celsiusstr. 1
Bremen 28359
Germany

Re: mSystems01287-22R1 (*In situ* cell division and mortality rates of SAR11, SAR86, *Bacteroidetes*, and *Aurantivirga* during phytoplankton blooms reveal differences in population controls)

Dear Dr. Bernhard M Fuchs:

Your manuscript has been accepted, and I am forwarding it to the ASM Journals Department for publication. For your reference, ASM Journals' address is given below. Before it can be scheduled for publication, your manuscript will be checked by the mSystems production staff to make sure that all elements meet the technical requirements for publication. They will contact you if anything needs to be revised before copyediting and production can begin. Otherwise, you will be notified when your proofs are ready to be viewed.

If you would like to submit a potential Featured Image, please email a file and a short legend to mSystems@asmusa.org. Please note that we can only consider images that (i) the authors created or own and (ii) have not been previously published. By submitting, you agree that the image can be used under the same terms as the published article. File requirements: square dimensions (4" x 4"), 300 dpi resolution, RGB colorspace, TIF file format.

We recognize that the video files can become quite large, and so to avoid quality loss ASM suggests sending the video file via <https://www.wetransfer.com/>. When you have a final version of the video and the still ready to share, please send it to mSystems staff at mSystems@asmusa.org.

Sincerely,

Jean-Baptiste Raina
Editor, mSystems

Journals Department
E-mail: mSystems@asmusa.org